# Knowledge, attitudes and practices toward Female Genital Schistosomiasis among women living in communities along the Volta Lake in Volta Region, Ghana

**Joyce Berkumwin Der**[ORCID]*, **Beatrice Efua Donkoh, Samuel Owusu Ansah, Alberta Kwakyewaa Bofa, Frank Oppong Kwafo**

Department of Epidemiology and Biostatistics, Fred N. Binka School of Public Health, University of Health and Allied Sciences, Hohoe, Volta Region, Ghana

* jdberkumwin@uhas.edu.gh

## Abstract

### Background

Female Genital Schistosomiasis (FGS) is a debilitating neglected tropical disease (NTD) caused by the parasite *Schistosoma haematobium*. In Ghana, FGS is endemic, especially in communities around Lake Volta. Despite the vulnerability of women in these communities, limited literature exists on their knowledge, attitudes, and practices (KAP). This study assessed KAP regarding FGS among women living in communities along the Volta Lake.

### Methods

A descriptive cross-sectional study was conducted using a multistage sampling technique to recruit women aged 18 years and above. Data were collected using a structured interviewer-administered questionnaire. Descriptive statistics were computed, and logistic regression analysis was used to identify factors associated with KAP. Variables with log-likelihood ratio p-value ≤0.2 in univariable analysis were included in multivariable models. Statistical significance was set at $p < 0.05$.

### Results

A total of 745 women were recruited. The mean age was 33 years (SD = 12.28), with the majority (36.4%) aged 21–30 years. Overall, 454 (60.9%) had poor knowledge of FGS, 446 (59.8%) had good attitudes, and 460 (61.7%) had poor practices. Women with good attitudes were significantly more likely to have good knowledge (aOR=4.67, 95% CI: 3.23–6.75, p<0.001). Those with secondary education were more likely to have good attitudes (aOR=2.28, 95% CI: 1.16–4.40, p=0.017) compared to those with no formal education. Women with good knowledge were also

**Data availability statement:** All relevant data are within the manuscript and its Supporting information files.

**Funding:** The author(s) received no specific funding for this work.

**Competing interests:** The authors have declared that no competing interests exist.

more likely to report good practices (aOR=1.42, 95% CI: 1.01–1.98, p = 0.041). Women residing in North Tongu were more likely to have good attitude (aOR=2.79, 95% CI: 1.68–4.63, p < 0.001) but poor practices (aOR=0.59, 95% CI: 0.36–0.97, p = 0.039).

## Conclusion

Despite relatively good attitudes, women in FGS-endemic communities around the Volta Lake showed poor knowledge and practices. These findings underscore the urgent need for targeted health education, awareness campaigns and integrating FGS into already existing sexual and reproductive health programs to improve community knowledge and practices. These will contribute toward achieving the 2030 NTD targets under Sustainable Development Goal 3.

## Author summary

Female Genital Schistosomiasis (FGS) is a neglected tropical disease caused by a waterborne parasite, *Schistosoma haematobium*. It affects millions of women in sub-Saharan Africa (SSA), including communities in Ghana near the Volta Lake. This study explored how much women in these communities know about FGS, their attitudes toward the disease, and the actions they take to prevent or manage it. Researchers surveyed 745 women aged 18 years and above and found that while many had a positive attitude toward addressing the disease, most had limited knowledge and poor health practices. Importantly, women with better knowledge tended to practice better prevention, and those with higher education were more likely to hold positive attitudes. These results show that knowledge and education are key factors influencing how women respond to FGS. This work is significant because it highlights gaps in health education in vulnerable populations. Without accurate knowledge and preventive practices, women remain at high risk of serious reproductive health complications. The findings support the need for targeted community health programs to educate women and promote better practices. This aligns with global health goals, including the United Nation's aim to eliminate NTDs by 2030 under Sustainable Development Goal 3.

## Introduction

Schistosomiasis is a disease of poverty that leads to chronic ill health. Infection is acquired when people come into contact with fresh water infested with the larval forms (cercariae) of parasitic blood flukes, known as schistosomes [1]. The infection, which affects both males and females, is prevalent in tropical and sub-tropical areas, in poor communities without potable water and adequate sanitation [1]. In endemic

regions, females are more at risk of infection. This is due to gender roles that expose them to frequent contact with the *Schistosoma* larvae excreted by the snail intermediate host. Domestic activities such as washing clothes, fetching water, and bathing make females more at risk of the infection [2].

Females can suffer gynecological complications from schistosomiasis known as Female Genital Schistosomiasis (FGS). It is a debilitating Neglected Tropical Disease (NTD) caused by the parasitic infection *Schistosoma haematobium*. An estimated 56 million women in Africa are currently suffering from FGS [1]. Women with FGS may experience symptoms such as vaginal discharge, bloody discharge, bleeding or pain during or after intercourse, and genital itching or burning [3]. If left untreated, FGS can gradually cause infertility or subfertility, premature birth, low birth weight, anemia, and menstrual disorders. These clinical manifestations are usually mistaken for Sexually Transmitted Infections (STIs) due to the similarities they share. Besides these adverse reproductive, emotional, and social consequences, FGS is also known to be a cofactor in acquiring human immunodeficiency virus type 1 (HIV-1), human papillomavirus (HPV), and cervical cancer [4]. This NTD is therefore poorly understood among women and often misdiagnosed by health workers [5].

Attitudes towards FGS are heavily influenced by sociocultural contexts and are often characterized by stigma, misconceptions, and gender-based inequalities. Evidence reveals that women with genital symptoms often experience shame and embarrassment, particularly when symptoms affect sexual relationships or fertility. This highlights how cultural taboos surrounding female reproductive health issues have created barriers to open discussion and care-seeking [6]. Current practices towards FGS have been heavily informed by how disease is understood. Practices related to FGS prevention, diagnosis, and management vary widely across endemic regions but are generally characterized by significant gaps and challenges. Prevention practices primarily focus on water contact behavior, as infection occurs through skin contact with freshwater containing infectious cercariae released by intermediate host snails [7]. However, women's water contact patterns are deeply intertwined with essential domestic responsibilities such as laundry, dishwashing, and bathing children. This has made behavioral change interventions particularly challenging without addressing structural water access issues [8].

In Ghana, FGS is endemic throughout the country with elevated infection levels in areas around Lake Volta [9]. Settlements along Lake Volta in Ghana have consistent water security challenges. Most of these settlements, mostly rural, experience a poor source of potable water. Poor water treatment plants, poor domestic water treatment practices, and disasters such as flood expose dwellers along the lake to water contaminated with *Schistosoma haematobium* by the freshwater snail [10]. A survey of the burden of FGS in the Volta Region indicated an alarming prevalence of 58.5% among the population sampled [11]. In Ghana, just like many other Sub-Saharan African countries, policies, control programs, and practices towards FGS is limited in health system. The disease is therefore underreported, underdiagnosed, and goes untreated until it worsens due to poor knowledge, awareness, and attitude of both health workers and victims [12].

Despite the Volta lake exposing communities along its banks to FGS and making them vulnerable to infection, there is limited literature on the KAP of women in these communities. Identifying KAP gaps in communities that largely depend on the Volta Lake for both domestic and commercial purposes will be imperative for FGS prevention. Addressing these gaps will also improve community attitude and practices towards infection and prevention. This study therefore assessed the knowledge, attitudes, and practices related to FGS among females living in communities along the Volta Lake of Ghana.

## Methods

### Ethics approval and consent to participate

Ethical approval to conduct the study was obtained from the Research Ethics Committee of the University of Health and Allied Sciences with approval number UHAS-REC A.10 [151] 23–24. The study was conducted in accordance with the principles of the Declaration of Helsinki. Permission was sought from the authorities of each community before the

commencement of the study. Written informed consent was obtained from all participants for their inclusion in the study. Contents of the study were fully disclosed to participants and their consent was sought after disclosure. Participants were informed that; their participation was voluntary. Participants were assured of confidentiality and anonymity, and that under no circumstance would their names and other details be linked to data analysis and dissemination of findings of the study.

## Study design

A descriptive cross-sectional study design was used to conduct the study. This ensured the collection of data from respondents at a point in time. The design was suitable for assessing the KAP and its associated factors among respondents.

## Study site

The study was conducted in three districts in the Volta Region in communities along the banks of the Volta Lake; South Tongu, North Tongu, and North Dayi. According to the 2021 population and housing census (PHS), South Tongu district has a population of 113,114 with 52, 488 males and 60,626 females. It has six [6] sub-districts and about 104 communities in the district. The North Tongu municipality on the other hand, has a population of 110,891 with 52,996 males and 57,895 females. It has 13 sub-districts and about 120 communities. While the North Dayi Municipality has a population of 39,268, with 19,075 males and 20,193 females. It has nine [9] sub-districts and over 60 communities. Fishing is the dominant occupation for the people in the communities in all the three districts [13].

## Sample size and study population

The sample size for this study was determined using Open Epi [14]. An assumed prevalence of 50% knowledge of FGS among women living in communities along the Volta Lake was used for the sample size calculation. Using Open Epi version 3 Source calculator, the sample size was calculated as follows;

$$\text{Sample size } n = \left[\text{DEF} * \text{Np}(1-p)\right] / \left[\left(d^2/Z^2_{1-\alpha/2} * (N-1) + p * (1-p)\right)\right]$$

Where Deff is the design effect (2), N is the finite population (138714), p is the proportion (50%), d is the precision (0.05) and z is the z-value at the 95% confidence level (1.96).

Based on the calculation above, the sample size was 767.

Using proportionate to size, the sample size for each district was calculated using the formula:

$$n = (\text{district population}/\text{total population size}) \text{ x total sample size}$$

Thus, the sample size for each district was South Tongu (335), North Tongu (320) and North Dayi (112).

The study population included females aged 18 years and above living in communities along the Volta Lake for at least six months however those who had lived in the community for less than six months were excluded from the study.

## Sampling method

A multi-stage sampling technique was used to select females from three districts out of 18 districts in the Volta region. The three districts were purposively selected because of the Volta Lake passing through them exposing communities along its banks to the risk of FGS. All sub-districts along the Volta Lake were purposively selected. Three communities in a sub-district were selected in each district using a simple random sampling method. Households within the communities were selected by locating the center of the community and spinning a bottle to identify the starting cluster. The nearest house in that cluster was selected to start the survey. After conducting interviews in the first house, moving in the right direction,

two houses were skipped and the third house selected as the next household. This procedure was repeated till the survey was completed in that community. Finally, females within a household who met the inclusion criteria were recruited as respondents for the study using the lottery method. By this, potential participants were made to select from two folded pieces of papers. Females who chose papers with "Yes" written on them were included in the study.

## Data collection tool

A semi-structured questionnaire designed using the KoboCollect toolbox was used to collect data for the study. The tool was designed by adapting knowledge and practice questions from the general knowledge of FGS [15]. Attitude questions were developed based on the socio-cultural context of beliefs and behavior toward FGS and other NTDs in the study districts. The tool was reviewed by the supervisor who is an expert in tropical diseases and the NTD focal persons of the Ghana Health Service in study districts. This was to assess the consistency and suitability of the tool to the context of the study participants. A face-to-face interview was conducted in the participants' house, and interviews lasted approximately 20–30 minutes. The instrument was designed in English under four main sections. The first section examined respondents' demographic characteristics while the second, third, and fourth sections assessed respondents' knowledge, attitudes, and practices respectively. During data collection, the questionnaire was administered in English and for participants who did not understand English, it was administered in a local language (Ewe or Twi). Data collectors were trained undergraduate public health students who could speak any of the local languages. The questionnaire was pretested among ten [10] females who were not part of the study to assess the flexibility of the data collection instrument and relevance of the questions to the study objectives.

## Study variables

The dependent variables in this study were knowledge, attitude, and practices, which were all composite variables. For knowledge, five sets of variables (ever heard of FGS, knowing how FGS is transmitted, knows about signs and symptoms, know FGS health implications, and knowing FGS affects women' quality of life) were used to generate the composite score, while for attitude and practices, six sets of variables were used to develop the composite scores. The set of questions that generated the attitude included FGS is not a serious health issue, relating well with FGS-infected people, women with FGS should be isolated, there is no need to seek care, willingness to support infected women, willingness to engage in FGS campaign, and seeking care when infected by FGS. Also, variables that assessed practices towards FGS included delayed in seeking care, sought care when infected, source of seeking medical care, practices/culture that influence infection, practicing existing beliefs about FGS, and swimming and wading in water. The independent variables in the study were mainly socio-demographic characteristics such as age, sex, marital status, religion, ethnicity among others.

## Data analysis

Data from the Kobo toolbox was downloaded as an MS Excel file and exported into STATA 17.0 for analysis. Descriptive statistics were performed at the univariable level and presented as frequencies, means, standard deviations, proportions, and percentages in tables. Knowledge, attitude, and practice scores were obtained by summing up responses to questions on the knowledge, attitude, and practices related to FGS to generate the composite outcome variables. Each right response attracted a score of 1, while a wrong response attracted 0. The mean was determined and used as a cut-off point for good or poor knowledge, attitude, or practice. Table 1 shows the mean scores used to categorize knowledge, attitude and practice into good or poor.

At the bi-variable level, the crude analysis test was used to determine the influence of the predicting factors on the generated outcome of knowledge, attitude, and practices. Univariable and multivariable binary logistic regression were

**Table 1. Mean score used to categorize knowledge, attitude and practices into good or poor.**

| Variable | Mean Score | Good | Poor |
|---|---|---|---|
| Knowledge | 2.66 | ≥2.66 | <2.66 |
| Attitude | 4.65 | ≥4.65 | <4.65 |
| Practice | 1.1 | ≥1.1 | <1.1 |

specified to estimate the crude odds ratios (cORs) and adjusted odds ratios (aORs) to assess the association between the outcome variable and its predictors. At the univariable analysis level, variables with log-likelihood p-value ≤ 0.2 and all other plausible variables were included in the multivariable model. This was done to avoid excluding variables that may not be significant in univariable analysis but could become significant or act as confounders once other variables were controlled for in the multivariable model [16]. At the multivariable level, a p-value of <0.05 was considered significant at a 95% confidence interval.

## Results

### Socio-demographic characteristics of respondents

Table 2 shows the distribution of sociodemographic characteristics of 745 respondents sampled for this study. The results show that majority of the respondents 271 (36.4%) were between the ages 21 – 30years with a mean age of 33 (12.28) years, while 351(47.1%) were married. Also, 718(96.4%) were Christians while 716(96.1%) were Ewes. About half of respondents 405(54.5%) had a family size of less than five while 424(56.9%) had up to basic level of education. Again, most study respondents 466(62.6%) were artisanal workers while 504(67.7%) receive monthly income less than 500 Ghana Cedis (38.04 USD using Xe currency converter accessed on 2nd May 2025).

**Knowledge on FGS.** The study explored the knowledge of respondents on FGS and found that 431(57.9%) had never heard of FGS, 467(62,7%) did not know the signs and symptoms of FGS, and 606(81.3%) did not know about the health consequences of FGS (S1 Table).

In exploring the overall knowledge of respondents, the study revealed that 454(60.9%) of the respondents had poor knowledge of FGS, while 291(39.1%) percent had good knowledge of FGS (Fig 1).

**Attitudes towards FGS.** This study also assessed the attitude of respondents towards FGS. It was revealed that 407(54.6%) did not identify FGS as a serious health problem while 495(66.4%) believed there was no need to seek care for FGS. Most respondents 685(91.9%) admitted they would relate well with FGS infected person, and 724(97.2%) were willing to support an FGS infected person (S2 Table).

Over all the study found 446(59.9%) of respondents had good attitude while 299(40.1%) had poor attitude towards FGS (Fig 2).

**Practices on FGS.** A total of 51 (6.9%) of respondents reported ever having FGS, of which 44 (86.7%) sought care for their symptoms, however, 33(75%) delayed in seeking care. Also, 32(72.7%) sought care from the health facility while 40(90.9%) were prompted to seek care due to the severity of the condition (S3 Table).

Overall, the study revealed 460(61.7%) had poor practices while 285(38.3%) had good practices (Fig 3).

### Factors influencing knowledge on FGS

The study examined the factors that influence knowledge on FGS. The findings from the crude logistic regression model reveal that respondents who had up to tertiary education were 2.57 times more likely to have good knowledge (cOR=2.57, [95%CI;1.22 – 5.37], p=0.012) compared to respondents who had no formal education. Also, artisans were 51 percent less likely to have good knowledge (cOR=0.49, [95%CI=0.27 – 0.87], p=0.016) compared to civil servants. The results also revealed that respondents with a good attitude towards FGS were 5.1 times more likely to have good knowledge (cOR=5.11,

**Table 2. Socio-demographic characteristics of respondents living in communities along the Volta Lake.**

| Variable | Frequency (N = 745) | Percentage |
|---|---|---|
| **Age (years)** | Mean (SD) = 33 (12.28) | |
| ≤ 20 | 91 | 12.2 |
| 21-30 | 271 | 36.4 |
| 31-40 | 192 | 25.8 |
| 41-50 | 119 | 15.9 |
| ≥ 51 | 71 | 9.5 |
| **Marital Status** | | |
| Married | 351 | 47.1 |
| Never Married | 342 | 45.9 |
| Divorced/widowed | 52 | 7.0 |
| **Religion** | | |
| Christian | 718 | 96.4 |
| Islamic | 22 | 2.9 |
| Traditional | 5 | 0.7 |
| **Ethnicity** | | |
| Akan | 8 | 1.1 |
| Ewe | 716 | 96.1 |
| Guan | 21 | 2.8 |
| **Family size** | | |
| < 5 | 405 | 54.4 |
| 5-10 | 327 | 43.9 |
| > 10 | 13 | 1.7 |
| **Level of education** | | |
| No formal education | 68 | 9.1 |
| Basic | 424 | 56.9 |
| Secondary | 200 | 26.9 |
| Tertiary | 53 | 7.1 |
| **Occupation** | | |
| Unemployed | 140 | 18.8 |
| Artisan | 466 | 62.6 |
| Student | 89 | 11.9 |
| Civil Servant | 50 | 6.7 |
| **Income (GHC)\*** | | |
| < 500 | 504 | 67.7 |
| 500-1000 | 172 | 23.1 |
| > 1000 | 69 | 9.2 |

*GHC=Ghana cedis

[95%CI = 3.61 − 7.23], p < 001). In addition, respondents who resided in South Tongu were 0.62 times less likely to have good knowledge on FGS compared to those who resided in North Dayi (cOR=0.62, [95%CI = 0.40 − 0.95], p = 0.030).

In the adjusted model, respondents with good attitude were still 4.67 times more likely to have good knowledge on FGS (aOR= 4.67, [95%CI = 3.23 − 6.75], p < 0.001) compared to respondents with poor attitude and those who resided in South Tongu were 0.57 still less likely to have good knowledge on FGS (aOR=0.57, [95%CI = 0.34 − 0.97], p = 0.038) (Table 3).

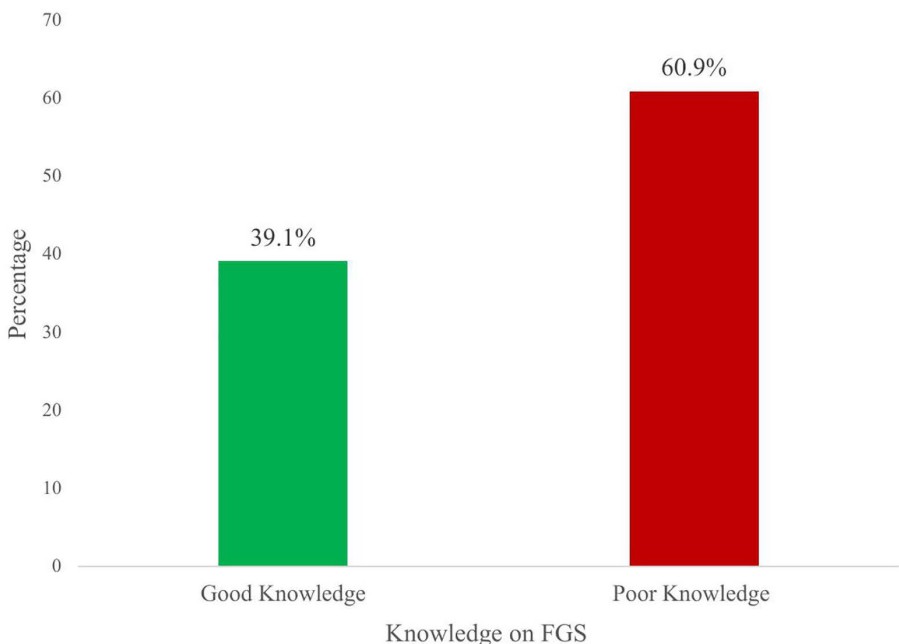

**Fig 1. Knowledge on FGS among women living in communities along the Volta Lake.**

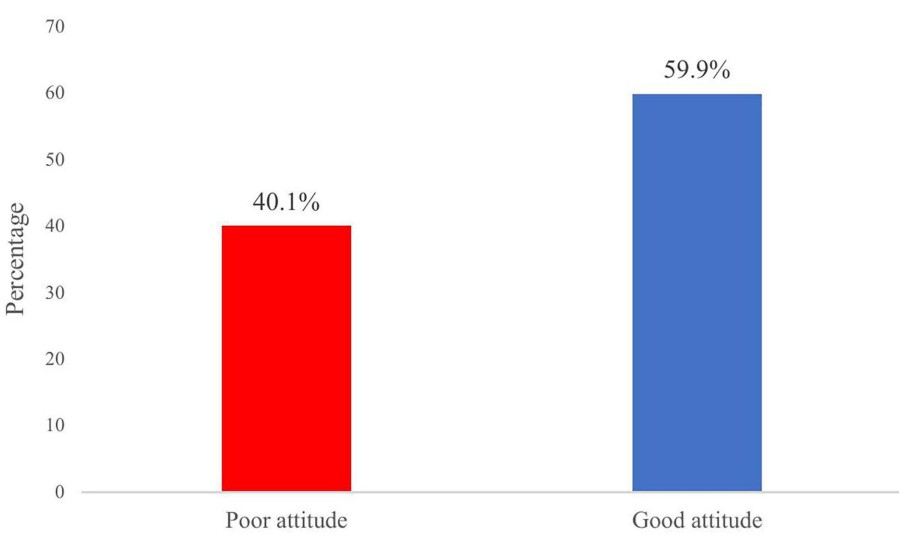

**Fig 2. Attitude towards FGS among women living in communities along the Volta Lake.**

## Factors influencing attitude towards FGS

The study found at the crude level that respondents with secondary (cOR=2.76, [95%CI = 1.58 − 4.85], p < 0.001) and tertiary education (cOR=2.71, [95%CI = 1.28 − 5.72], p = 0.009) were more likely to have good attitude compared to those with no education. Also, those with good knowledge were 5.11 times more likely to have a good attitude compared to

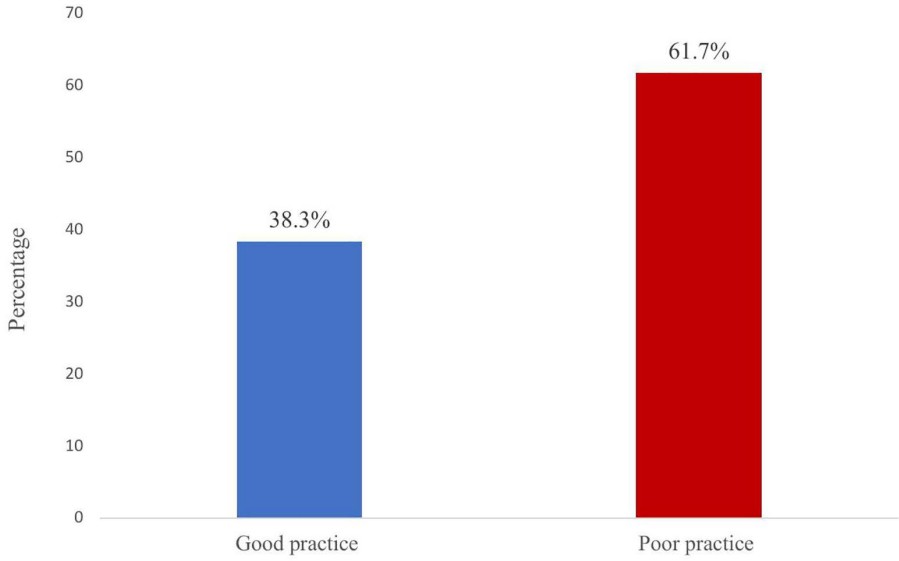

**Fig 3. Practices towards FGS among women living in communities along the Volta Lake.**

those with poor knowledge (cOR=5.11, [95%CI = 3.61 – 7.23], p < 0.001) and those who resided in North Tongu were 3.15 times more likely to have good attitude compared to those in North Dayi (cOR=3.15, [95%CI = 32.02 – 4.92], p < 0.001). In the adjusted model, those with secondary education remained 2.28 times more likely to have a good attitude (aOR=2.28, [95%CI = 1.16 – 4.40], p = 0.017) compared to individuals with no formal education. Again, respondents who had good knowledge were 4.86 times more likely to have good attitude towards FGS (aOR=4.86, [95%CI = 3.37 – 7.01], p < 0.001) compared to those with poor knowledge of FGS and those who resided in North Tongu were 2.79 times more likely to have good attitude (aOR=2.79, [95%CI = 1.68 – 4.63], p < 0.001) compared to those in North Dayi as shown in Table 4.

### Factors influencing practices

The study also explored the factors that influence practices towards FGS. The results revealed that at the crude level, individuals within the ages 41–50 years were 2.15 times more likely to have good practices towards FGS (cOR=2.15, [95%CI = 1.29 – 4.61], p = 0.008) compared to individuals ≤20 years. Also, individuals ≥50 years were 2.4 times more likely to have good practices towards FGS (cOR=2.43, [95%CI = 1.29 – 4.61], p = 0.006) compared to individuals ≤20years. Respondents residing in North Tongu were 0.42 times less likely to have good practices on FGS (cOR=0.42, [95%CI = 0.27 – 0.66], p < 0.001) compared to those in North Dayi.

In the adjusted model, respondents with family size <5 were 4.64 times more likely to have good practices (aOR=4.64, [95%CI = 1.14 – 18.94], p = 0.032) compared to a family size of ≥10. Also, respondents who had good knowledge were 1.42 times more likely to have good practices (aOR=1.42, [95%CI = 1.01 – 1.98], p = 0.041) compared to those with poor knowledge and those residing in North Tongu were 0.59 times less likely to have good practices (aOR=0.59, [95%CI = 0.36 – 0.97], p = 0.039) compared to those in North Dayi (Table 5).

### Discussion

This study assessed the KAP and associated factors toward FGS among females living in communities along the Volta Lake in the Volta Region. The study found poor knowledge, good attitude and poor practices among females in the study communities.

**Table 3. Factors influencing knowledge on FGS among women living in communities along the Volta Lake.**

| Variable | cOR (95% CI), p-value | aOR (95% CI), p-value |
|---|---|---|
| **Age** | **0.210*** | |
| ≤20 | Ref | Ref |
| 21-30 | 1.47 (0.89–2.43), 0.130 | 1.33 (0.66–2.67), 0.419 |
| 31-40 | 1.14 (0.67–1.93), 0.628 | 1.13 (0.53–2.41), 0.759 |
| 41-50 | 1.65 (0.93–2.91), 0.084 | 1.93 (0.86–4.34), 0.110 |
| >50 | 1.74 (0.92–3.29), 0.088 | 1.78 (0.72–4.40), 0.213 |
| **Marital Status** | 0.695* | |
| Divorced | Ref | |
| Married | 0.78 (0.433–1.39), 0.395 | |
| Never Married | 0.81 (0.45–1.45), 0.479 | |
| **Religion** | **0.227*** | |
| Traditional | Ref | Ref |
| Christian | 7.24 (0.39–13.51), 0.181 | 7.23 (0.32–164.14), 0.214 |
| Islamic | 4.33 (0.21–90.04), 0.344 | 3.48 (0.13–91.69), 0.455 |
| **Ethnicity** | 0.714* | |
| Guan | Ref | |
| Akan | 1.93 (0.390–9.35), 0.412 | |
| Ewe | 1.24 (0.50–3.04), 0.634 | |
| **Family size** | **0.096*** | |
| Above 10 | Ref | Ref |
| <5 | 0.85 (0.29–2.54), 0.777 | 1.01 (0.20–3.42), 0.993 |
| 5-10 | 1.19 (0.30–3.55), 0.759 | 1.08 (0.31–3.64), 0.917 |
| **Level of education** | **<0.001*** | |
| No formal education | Ref | Ref |
| Basic | 0.66 (0.39–1.11), 0.118 | 0.62 (0.33–1.16), 0.134 |
| Secondary | 1.03 (0.59–1.79), 0.910 | 0.82 (0.40–1.67), 0.582 |
| Tertiary[NS] | **2.57 (1.22–5.37), 0.012** | 2.49 (0.87–7.19), 0.090 |
| **Occupation** | **0.009*** | |
| Civil Servant | Ref | Ref |
| Artisan[NS] | **0.49 (0.27–0.87), 0.016** | 1.51 (0.62–3.66), 0.359 |
| Student | 0.79 (0.30–1.57), 0.504 | 1.78 (0.69–4.59), 0.231 |
| Unemployed | 0.76 (0.30–1.44), 0.396 | 2.02 (0.81–5.03), 0.129 |
| **Income** | 0.181* | |
| Above 1000 | Ref | Ref |
| <500 | 0.64 (0.39–1.06), 0.085 | 0.95 (0.49–1.84), 0.877 |
| 500-1000 | 0.77 (0.44–1.34), 0.353 | 1.33 (0.63–2.79), 0.450 |
| **Attitude** | **<0.001*** | |
| Poor attitude | Ref | Ref |
| Good attitude | **5.11 (3.61–7.23), <0.001** | **4.67 (3.23–6.75), <0.001** |
| **Practices** | **0.023*** | |
| Poor practices | Ref | Ref |
| Good practices | 1.20 (0.88–1.622), 0.235 | 1.42 (0.90–2.02)0.051 |

*(Continued)*

**Table 3.** (Continued)

| Variable | cOR (95% CI), p-value | aOR (95% CI), p-value |
|---|---|---|
| **District** | **0.001*** | |
| North Dayi | Ref | Ref |
| North Tongu | 1.42(0.93–2.19), 0.106 | 1.00 (0.58–1.74), 0.99 |
| South Tongu | **0.62(0.40–0.95), 0.030** | **0.57 (0.34–0.97), 0.038** |

(*) = log likelihood probability value; significant at ≤0.2, NS= Not significant at the adjusted level

## Poor knowledge and its public health implications

The findings revealed that 60.9 percent of the respondents had poor knowledge of FGS consistent with other studies across sub-Saharan Africa (SSA) that reported similarly low awareness [4,17]. This study's finding also aligns with a study conducted by Manyeh et al. (2024), in Ghana who also reported 50.3 percent poor knowledge on FGS [18]. The differences in the poor knowledge margins may be attributed to the peri-urban nature of Manyeh and colleagues' study site. People living in peri-urban areas have access to health education compared to people in rural communities [19]. This poor knowledge underscores the persistent global neglect of FGS within community health education and highlights the broader challenge of addressing NTDs that disproportionately affect females in rural communities [20]. Poor knowledge levels also reflect a critical barrier to early diagnosis, treatment, and prevention of FGS.

The consequences of delayed diagnosis and inappropriate treatment perpetuate transmission within endemic areas. Similar patterns have been documented in other SSA contexts, including Tanzania and Madagascar [17,21], suggesting that FGS remains invisible within both community and healthcare systems.

The gendered nature of FGS introduces additional barriers to knowledge acquisition. Female genital health conditions often carry social stigma, limiting open discussion and education. This stigma may contribute to persistent knowledge gaps, particularly in communities with conservative attitudes toward discussing reproductive health.

The study further shows that knowledge is, however, influenced by attitude and occupation. People who have a good attitude positively influenced knowledge of FGS. As a disease with stigma [6], people who show good attitude towards the disease transmission, prevention, treatment, and support to victims may want to better understand the infection to be more empowered to exhibit their good attitude. This may influence their knowledge gains regarding the condition. Similar findings have been reported in Cameroon, where individuals with supportive views toward affected persons also showed higher levels of awareness [22]. This points to an opportunity for interventions to leverage positive community attitudes as a foundation for strengthening knowledge on FGS.

## Linking community KAP gaps to healthcare worker knowledge deficits

An important but often overlooked determinant of community KAP is the knowledge and competence of healthcare workers (HCWs). Evidence from Cameroon and Tanzania shows that many HCWs are unable to correctly diagnose or manage FGS and often misclassify it as a sexually transmitted infection [22,23]. In Ghana, studies among final-year health trainees also revealed poor awareness and clinical understanding of FGS [24]. Further evidence from Ghana reveals that before FGS training interventions, HCW awareness was below 8 percent; after training, awareness rose to over 61 percent [25]. This lack of HCW capacity limits health education outreach and reduces the likelihood of accurate counselling, thereby perpetuating misinformation within communities. Strengthening HCW training on FGS is therefore essential. Continuous professional development and inclusion of FGS modules in Sexual and Reproductive Health (SRH) and NTD curricula could bridge this systemic gap and improve community-level awareness through trusted health educators.

**Table 4. Factors influencing attitude towards FGS among women living in communities along the Volta Lake.**

| Variable | cOR (95% CI), p-value | aOR (95% CI), p-value |
|---|---|---|
| **Age** | 0.584* | |
| ≤20 | Ref | |
| 21-30 | 1.32 (0.82–2.12), 0.258 | |
| 31-40 | 1.34 (0.81–2.21), 0.259 | |
| 41-50 | 1.25 (0.72–2.17), 0.419 | |
| ≤50 | 0.94 (0.51–1.75), 0.856 | |
| **Marital Status** | 0.188* | |
| Divorced | Ref | Ref |
| Never Married | 1.67 (0.93–2.98), 0.084 | 1.15 (0.58–2.31), 0.685 |
| Married | 1.42 (0.70–2.53), 0.236 | 1.06 (0.54–2.09), 0.866 |
| **Religion** | 0.269* | |
| Traditional | Ref | Ref |
| Christian | 4.50 (0.71–28.74), 0.112 | 1.15 (0.22–10.48), 0.676 |
| Islamic | 5.12 (0.67–39.07), 0.115 | 1.24 (0.14–10.59), 0.844 |
| **Ethnicity** | 0.986* | |
| Guan | Ref | |
| Ewe | 0.94 (0.39–2.23), 0.880 | |
| Akan | 0.99 (0.20–4.86), 0.990 | |
| **Family size** | 0.005* | |
| >10 | Ref | Ref |
| <5 | 0.78 (0.26 -2.31), 0.650 | 0.61 (0.19–1.99), 0.413 |
| 5-10 | 1.27 (0.43–3.81), 0.666 | 0.69 (0.21–2.29), 0.544 |
| **Level of education** | <0.001* | |
| No formal education | Ref | Ref |
| Basic | 1.52 (0.91–2.54), 0.107 | **1.83 (1.03–3.30), 0.045** |
| Secondary | **2.76 (1.58–4.85), <0.001** | **2.28 (1.16–4.40), 0.017** |
| Tertiary[NS] | **2.71 (1.28–5.72), 0.009** | 2.11 (0.39–3.16), 0.836 |
| **Occupation** | 0.015* | |
| Civil Servant | Ref | Ref |
| Artisan | 0.54 (0.29–1.01), 0.054 | 0.72(0.20–1.76), 0.474 |
| Student | 0.94 (0.45–1.98), 0.875 | 0.74(0.29–1.88), 0.474 |
| Unemployed | 0.83 (0.42–1.66), 0.603 | 0.80 (0.32–2.04), 0.648 |
| **Income** | 0.677* | |
| Above 1000 | Ref | |
| <500 | 0.99 (0.50–1.67), 0.997 | |
| 500-1000 | 0.86 (0.49–1.51), 0.589 | |
| **Knowledge** | <0.001* | |
| Poor knowledge | Ref | Ref |
| Good knowledge | **5.11 (3.61–7.23), <0.001** | **4.86 (3.37–7.01), <0.001** |

*(Continued)*

**Table 4.** (Continued)

| Variable | cOR (95% CI), p-value | aOR (95% CI), p-value |
|---|---|---|
| **Practices** | 0.308* | |
| Poor practices | Ref | |
| Good practices | 0.86 (0.63–1.15), 0.308 | |
| District | **0.01*** | |
| North Dayi | Ref | Ref |
| North Tongu | **3.15(2.02–4.92) <0.001** | **2.79(1.68–4.63), <0.001** |
| South Tongu | 1.03(0.68–1.57)0.871 | 1.19(0.71–1.91), 0.461 |

(*) = log likelihood probability value; significant at ≤0.2, NS= Not significant at the adjusted level.

## Attitudes toward FGS and the role of education

The study found a high level of good attitude towards FGS which highlights the willingness of participants to provide the needed support to FGS victims, engage in FGS campaigns, and also seek care when infected with FGS. This aligns with a similar study conducted in Ghana that also reported good attitude towards FGS [26]. Public health officials and stakeholders should therefore capitalize on the good attitude in these communities to improve upon community members' knowledge and awareness of the disease's causes, preventive measures, and better health-seeking behaviors.

Although, majority of participants demonstrated good attitudes toward FGS, nearly 40 percent had poor attitude. This substantial minority with poor attitudes represents a barrier to effective preventive practices, better treatment behaviors, and support for victims. However, education was significantly associated with more positive attitudes where respondents with formal education were more likely to express good attitudes compared to those without formal education. This aligns with findings from Ghana and Madagascar, where educational interventions improved willingness to seek care and support affected women [21]. These findings suggest that education serves as both a protective and enabling factor for health-seeking behaviour. However, positive attitudes alone do not guarantee appropriate practices; they must be coupled with accurate knowledge and access to preventive and curative services.

## Gaps in practices and structural barriers

Despite good attitudes, most respondents (61.7%) exhibited poor practices toward FGS prevention and management signaling a major implementation gap. This gap in practice may be due to the poor knowledge found in this study. Poor understanding of FGS may contribute to engaging in risky behaviors without assessing the dangers of getting infected. Similar findings have been reported in DR Congo and Ethiopia [4,27]. Inadequate health infrastructure, continued reliance on unsafe water sources, and cultural stigma associated with genital health could be accounting for these poor practices.

Improving practices requires both behavioural change communication and structural interventions, particularly access to clean water and safe sanitation. In this study, knowledge was found to influence practices. Influence of knowledge on practices towards FGS has been documented by several studies [4,18,24,27]. Therefore, strategies such as integrating FGS prevention and education into various reproductive health and NTD programs running within the health system could improve knowledge and adherence to practices that could prevent FGS transmission.

## Integrating FGS into sexual and reproductive health services

Given its clinical overlap with gynaecological conditions (e.g., bleeding, discharge etc), FGS should be repositioned as a key SRH concern. Integration within existing SRH platforms such as cervical cancer screening, HIV counselling, sexually transmitted infection (STI) clinics and family planning programs presents opportunities for dual benefit: women already attending clinics for cervical

PLOS Neglected Tropical Diseases

**Table 5. Factors influencing practices towards FGS among women living in communities along the Volta Lake.**

| Variable | cOR (95% CI), p-value | aOR (95% CI), p-value |
|---|---|---|
| **Age** | **<0.001*** | |
| ≤20 | Ref | Ref |
| 21-30 | 0.91 (0.54–1.51), 0.705 | 0.69 (0.36–1.31), 0.130 |
| 31-40 | 1.42 (0.84–2.40), 0.188 | 0.94 (0.47–1.88), 0.871 |
| 41–50[NS] | **2.15 (1.22–3.79), 0.008** | 1.31 (0.63–2.73), 0.470 |
| ≤50[NS] | **2.43 (1.29–4.61), 0.006** | 1.67 (0.72–3.85), 0.231 |
| **Marital Status** | 0.782* | |
| Divorced | Ref | |
| Never Married | 0.81 (0.45–1.46), 0.487 | |
| Married | 0.85 (0.47–1.52), 0.582 | |
| **Religion** | **0.038*** | |
| Traditional | Ref | Ref |
| Christian | 7.12(0.39–129.24), 0.184 | 15.41(0.69–343.12), 0.084 |
| Islamic | 1.97 (0.09–44.29), 0.668 | 5.35(0.19–152.15), 0.326 |
| **Ethnicity** | **0.189*** | |
| Guan | Ref | Ref |
| Akan | 3.88 (0.73–20.61), 0.110 | 2.79 (0.43–17.80), 0.280 |
| Ewe | 2.46 (0.86–6.99), 0.093 | 0.79 (0.19–3.18), 0.735 |
| **Family size** | **<0.001*** | |
| >10 | Ref | Ref |
| <5 | 3.91 (0.98–15.56), 0.053 | **4.64(1.14–18.94), 0.032** |
| 5-10 | 1.95 (0.49–7.70), 0.347 | 2.67(0.65–10.97), 0.172 |
| **Level of education** | **0.027*** | |
| No formal education | Ref | Ref |
| Basic | 0.97 (0.58–1.62), 0.906 | 1.11(0.60–2.03), 0.740 |
| Secondary | 0.63 (0.36–1.11), 0.110 | 1.22(0.61–2.44), 0.568 |
| Tertiary | 0.49 (0.23–1.06), 0.070 | 0.840(0.38–3.32), 0.840 |
| **Occupation** | **<0.001*** | |
| Civil Servant | Ref | Ref |
| Artisan[NS] | **1.88 (1.01–3.51), 0.047** | 1.60 (0.66–2.03), 0.300 |
| Student | 0.72 (0.33–1.55), 0.400 | 0.80 (0.31–2.09), 0.648 |
| Unemployed | 0.89 (0.44–1.70), 0.748 | 0.86 (0.34–2.10), 0.751 |
| **Income** | 0.529* | |
| >1000 | Ref | |
| <500 | 1.33 (0.78–2.27), 0.290 | |
| 500-1000 | 1.38 (0.77–2.48), 0.278 | |
| **Attitude** | 0.308* | |
| Poor attitude | Ref | |
| Good attitude | 0.86 (0.63–1.15), 0.308 | |

*(Continued)*

**Table 5.** (Continued)

| Variable | cOR (95% CI), p-value | aOR (95% CI), p-value |
|---|---|---|
| **Knowledge** | **0.235*** | |
| Poor knowledge | Ref | Ref |
| Good knowledge | 1.20 (0.88–1.62), 0.235 | **1.42(1.01–1.98), 0.041** |
| District | **<0.001*** | |
| North Dayi | Ref | Ref |
| North Tongu | **0.42(0.27–0.66) <0.001** | **0.59(0.36–0.97), 0.039** |
| South Tongu | 1.16(0.76–1.75), 0.495 | 1.03(0.64–1.65), 0.907 |

(*) = log likelihood probability value; significant at ≤0.2, NS= Not significant at the adjusted level

cancer screening or STI management could simultaneously receive FGS screening and education. The World Health Organization and partners have emphasized this integration as a cost-effective strategy to enhance detection and reduce stigma [28,29]. The "Minimum service package for the integration of FGS into SRH interventions" [30] provides a feasible model that could be adapted to the Ghanaian context. Such integration ensures comprehensive, stigma-free care and maximizes resource efficiency.

## Strengths and limitations

This study provides important baseline evidence on community knowledge, attitudes, and practices toward FGS in endemic areas of Ghana. By focusing on women living along the Volta Lake one of the country's highest-risk areas, the findings are highly relevant for public health planning and interventions. The study highlights critical gaps that can inform policy, health education and the integration of FGS into broader reproductive health services.

Nonetheless, several limitations must be acknowledged. The cross-sectional design prevents causal inference and the use of the bottle spinning technique in selecting households could introduce selection bias where the households chosen along that line may not represent the entire community. Reliance on self-reported data raises the risk of recall and social desirability bias. Future research should examine health system related factors influencing diagnosis, treatment and prevention of FGS to provide more robust evidence to guide comprehensive strategies to improve FGS prevention and management.

## Conclusion

Women in FGS-endemic communities along the Volta Lake had poor knowledge and practices but generally positive attitudes toward FGS. Understanding of disease transmission, symptoms, and appropriate health-seeking behaviors was limited. Education and attitude significantly influenced knowledge and practices, underscoring the multifaceted determinants of FGS-related behavior.

These findings highlight the urgent need for targeted interventions to improve awareness and promote effective preventive practices among women in endemic areas. Strengthening knowledge and community-level practices; integrating FGS education, screening and prevention into existing SRH and NTD programs; training of HCWs to enhance diagnostic capacity; and community engagements are not only critical to reduce local transmission but also to advance Ghana's progress toward Sustainable Development Goal 3, particularly the target of ending NTD epidemics by 2030. To achieve and scale up these interventions will require support from international organizations in collaboration with the Ghana Health Service, Non-governmental Organizations and community leaders.

## Supporting information

**S1 Table. Knowledge of FGS among women living in communities along the Volta Lake.**
(DOCX)

**S2 Table. Attitude towards FGS among women living in communities along the Volta Lake.**
(DOCX)

**S3 Table. Practices towards FGS among women living in communities along the Volta Lake.**
(DOCX)

**S4 Table. Minimum dataset.**
(XLSX)

## Acknowledgments

The researchers thank all the women who participated in the study for their cooperation and to all the community leaders who granted permission for the study in their communities.

## Author contributions

**Conceptualization:** Joyce Berkumwin Der, Beatrice Efua Donkoh, Samuel Owusu Ansah, Alberta Kwakyewaa Bofa.

**Data curation:** Beatrice Efua Donkoh, Samuel Owusu Ansah, Alberta Kwakyewaa Bofa.

**Formal analysis:** Joyce Berkumwin Der, Frank Oppong Kwafo.

**Investigation:** Joyce Berkumwin Der, Beatrice Efua Donkoh, Samuel Owusu Ansah, Alberta Kwakyewaa Bofa.

**Methodology:** Beatrice Efua Donkoh, Samuel Owusu Ansah, Alberta Kwakyewaa Bofa.

**Supervision:** Joyce Berkumwin Der.

**Validation:** Joyce Berkumwin Der.

**Writing – original draft:** Frank Oppong Kwafo.

**Writing – review & editing:** Joyce Berkumwin Der, Beatrice Efua Donkoh, Samuel Owusu Ansah, Alberta Kwakyewaa Bofa, Frank Oppong Kwafo.

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
