## [Decision Letter · Decision Letter 0]

20 Aug 2025

PNTD-D-25-01109

Knowledge, attitudes and practices toward Female Genital Schistosomiasis among women living in communities along the Volta Lake in Volta Region, Ghana

Dear Dr. DER,

Thank you for submitting your manuscript to PLOS Neglected Tropical Diseases. After careful consideration, we feel that it has merit but does not fully meet PLOS Neglected Tropical Diseases's publication criteria as it currently stands. Therefore, we invite you to submit a revised version of the manuscript that addresses the points raised during the review process.

Please submit your revised manuscript within 60 days, by October 18th. If you will need more time than this to complete your revisions, please reply to this message or contact the journal office at plosntds@plos.org. Please include the following items when submitting your revised manuscript:

We look forward to receiving your revised manuscript.

Kind regards,

Angela Monica Ionica, Ph.D.

Academic Editor

Jong-Yil Chai

Section Editor

Shaden Kamhawi

co-Editor-in-Chief

Paul Brindley

co-Editor-in-Chief

**Journal Requirements:**

1) Please upload all main figures as separate Figure files in .tif or .eps format. For more information about how to convert and format your figure files please see our guidelines: 

2) We notice that your supplementary Tables are included in the manuscript file. Please remove them and upload them with the file type 'Supporting Information'. Please ensure that each Supporting Information file has a legend listed in the manuscript after the references list.

3) We note that your Data Availability Statement is currently as follows: "All relevant data are within the manuscript and its Supporting Information files". Please confirm at this time whether or not your submission contains all raw data required to replicate the results of your study. Authors must share the “minimal data set” for their submission. PLOS defines the minimal data set to consist of the data required to replicate all study findings reported in the article, as well as related metadata and methods (https://journals.plos.org/plosone/s/data-availability#loc-minimal-data-set-definition).

4) As required by our policy on Data Availability, please ensure your manuscript or supplementary information includes the following:

**Reviewers' Comments:**

Reviewer's Responses to Questions

**Key Review Criteria Required for Acceptance?**

**Methods**

-Are the objectives of the study clearly articulated with a clear testable hypothesis stated?

-Is the study design appropriate to address the stated objectives?

-Is the population clearly described and appropriate for the hypothesis being tested?

-Is the sample size sufficient to ensure adequate power to address the hypothesis being tested?

-Were correct statistical analysis used to support conclusions?

-Are there concerns about ethical or regulatory requirements being met?

Reviewer #1: Objective clear, appropriate design and analysis, no ethical concerns

Reviewer #2: **Major Revision**

The objectives of the study are clearly articulated, and the study design is appropriate for addressing these objectives. There are no concerns regarding ethical or regulatory requirements. However, please consider the following suggestions:

** Data Collection Tools”

The manuscript does not sufficiently describe how the Knowledge, Attitudes, and Practices (KAP) variables were defined or validated, which affects reproducibility and reliability. The composite scores for KAP were constructed from selected questions. However, they do not reference to existing KAP frameworks for FGS or similar NTDs. Additionally, there is no justification for why certain variables were included while others were excluded. The questionnaire was adapted but does not describe (i) the source literature for the "general knowledge of FGS" questions (Line 166) and (ii) whether content validity (expert review) or internal consistency was assessed. There is no mention of how translation to local language adaptation, which is critical for KAP surveys. Minor revisions can address this. Adding 1–2 sentences on tool development and validation would strengthen the study’s credibility.

** Data Analysis**

-Line 202. Please develop a justification for the use of a p-value threshold of ≤ 0.2 for selecting variables in bivariate analysis before multivariable regression. It is unconventional and needs to be better explained.

**Address as limitations**:

-The authors have opted to use “the bottle-spinning method: for household selection. This could introduce site-specific bias. Briefly acknowledge this in the limitations.

Reviewer #3: (No Response)

**Results**

-Does the analysis presented match the analysis plan?

-Are the results clearly and completely presented?

-Are the figures (Tables, Images) of sufficient quality for clarity?

Reviewer #1: Yes

Reviewer #2: **Minor Revision**

The analysis presented aligns with the analysis plan, and the results are clearly displayed. I suggest some improvements in data presentations and justifications.

1. Revision of Table 1:

Improve the readability of Table 1 by clarifying income reporting. Please specify the currency conversion date (line 225: 500 ghs = 38.04 usd). Cite what type of exchange rates fluctuate (cite the source/year). Marital status. Were there no widows, or were you grouped with the divorced? Please clarify.

2. Explain the p < 0.2 Rationale for the regression results.

3. Add a table on the methods that clarifies what determines what was “Good” or “poor” knowledge, attitude and practice, ideally linked to a framework and a validated tool. Help the reader and remind them in the narrative of the results.

4. Tables and figures support the text; however, some formatting refinements are needed, especially in the tables. I recommend re-editing all of them to improve readability. Here are some suggestions: Highlight Significance: Bold significant results (e.g., p<0.05). Merge "P-Value" Column: Remove the standalone column; keep p-values with cOR/aOR. Align Reference Categories: Place "Ref" under both cOR and aOR columns for consistency. Group Non-Significant Variables: Use footnotes (for instance, "NS = Not significant in adjusted model").

5. Why did the authors choose not to assess differences in sites? I recommend adding site stratified results (even briefly) to strengthen the analysis. If practices vary by district, interventions should then be tailored. For instance, a particular site may need more awareness campaigns. If authors choose not to proceed with the suggestion, please add this as a limitation or need for future research.

Reviewer #3: (No Response)

**Conclusions**

-Are the conclusions supported by the data presented?

-Are the limitations of analysis clearly described?

-Do the authors discuss how these data can be helpful to advance our understanding of the topic under study?

-Is public health relevance addressed?

Reviewer #1: Yes

Reviewer #2: **Major Revision**

Your discussion provides a solid foundation for improving the management of schistosomiasis and FGS in Ghana. The discussion has a strong foundation but requires tighter logic, broader evidence integration, and policy relevant specifics. Please consider the following suggestions:

1. The discussion jumps between findings without a narrative. Adding subtitles to the main arguments and points of discussion that the authors wish to convey will improve readability and help organise the structure.

2. The discussion fails to address how poor HCW awareness perpetuates community KAPs . HCWs are frontline educators. Their lack of training in FGS diagnosis and management directly impacts community KAP (there is a lot of studies on this). This omission undermines the call for systemic improvements. While NTD programs are mentioned, the discussion does not explicitly tie FGS to SRH services, despite its clinical overlap with gynaecological conditions. SRH platforms (like cervical cancer screening) could amplify FGS detection and education. The need for integration can be found in some key studies: (i) Integration of prevention and control measures for female genital schistosomiasis, HIV and cervical cancer. (ii) Human rights as a framework for eliminating female genital schistosomiasis. (iii) Minimum service package for the integration of Female Genital Schistosomiasis into sexual and reproductive health and rights interventions. (iv) Urogenital schistosomiasis (UGS) and female genital schistosomiasis (FGS) in Cameroon: an observational assessment of key reproductive health determinants of girls and women in the Matta Health Are. (v) “We know about schistosomiasis but we know nothing about FGS”: A qualitative assessment of knowledge gaps about female genital schistosomiasis among communities living in Schistosoma haematobium endemic districts of Zanzibar and Northwestern Tanzania. (vi) “Female genital schistosomiasis is a sexually transmitted disease”: Gaps in healthcare workers’ knowledge about female genital schistosomiasis in Tanzania.

-Please consider including a list of the community KAP gaps to HCW training deficits and propose integration with SRH services.

3. Strengthen the discussion of your findings with he previous work of Mazigo’s in Tanzania and the FAST package in Ghana and Madagascar (some were listed above). These gaps miss opportunities to align findings with proven strategies for KAP improvement.

4. Strengthen your limitation chapter. Address methodological limitations like the cross-sectional design, reliance on self-reported data without triangulation of data. The lack of validation on the KAP tools. Reflect on the missed opportunities of the design of the study on examining healthcare system barriers, propose future research directions in this vein.

5. Add a conclusion that summarises specific recommendations for policymakers, international organisations, and local stakeholders to support the community awareness and future research on FGS community awareness.

Reviewer #3: (No Response)

**Editorial and Data Presentation Modifications?**

Reviewer #1: No

Reviewer #2: -Formatting: "Schistosoma haematobium" should be italicized throughout. Eliminate redundant terms (like multiple uses of "majority"). Correct: "whiles" to "while"Correct: "crudes level" to "crude level".

-Citation Consistency: The manuscript currently mixes APA (author-date) and Vancouver (numbered) citation styles. Please standardize to one format throughout text and references.

Reviewer #3: (No Response)

**Summary and General Comments**

Reviewer #1: At Line 296 : ", respondents with family size >5 were times more likely ..." it should be less than 5, right ?

Reviewer #2: *Major Revision*

This study provides important insights into knowledge, attitudes and practices (KAP) regarding female genital schistosomiasis (FGS) among vulnerable populations in Ghana. While the research effectively identifies critical gaps in health education, major revisions are needed to enhance its scientific rigor and policy relevance. Key areas requiring attention include: (i)methodological limitations: The use of unvalidated KAP tools and potential sampling biases should be explicitly addressed, with justification provided for the chosen methodology; (ii) the discussion should more thoroughly examine how healthcare worker knowledge gaps links with community level misunderstandings about FGS, and how this affects health seeking behaviors; (iii) the paper would benefit from specific recommendations on integrating FGS education into existing sexual and reproductive health services; (iv) claims should be supported by appropriate references to comparable studies in similar settings, with particular attention to West African research on neglected tropical diseases.

Introduction

Lines 81-82): "The statement '112 million people worldwide are infected with FGS' requires clarification. Given limited global burden studies, I recommend revising to: 'An estimated 56 million women in sub-Saharan Africa currently suffer from FGS (1).' This maintains accuracy while reflecting available evidence."

Lines 91-92: you state that “The problem is further worsened by social stigma and destruction of the hymen, leading to accusations of sexual promiscuity”. This claim needs stronger evidence. Reference 6 (Kukula) doesn't support this specific claim. Please either remove this statement, or cite a more appropriate reference documenting this phenomenon.

Reviewer #3: The manuscript discusses a significant and relevant subject on neglected tropical diseases (NTDs). However, it needs to be revised before acceptance. Please consider the following comments to improve the quality of the manuscript.

Comment: #1

(Sampling Method)

How many districts on the Volta region? Why were these three districts chosen based on what specific criteria? It specifies that the first household was sampled using a bottle spin, but not how other households were selected.

Comment: #2

(Discussion)

• In the discussion section, there are statements about KAP (e.g. "knowledge is affected by attitude and occupation," "the education increases the good attitude by 2.6 times") without having presented any of the required statistical evidence to support them. For all such claims, the manuscript must contain the estimates of the association (e.g., adjusted odds ratio) and 95% confidence intervals, and P-values from multivariable logistic analysis. This information would allow readers to assess the relationship.

• The authors used different reference styles in the discussion.

PLOS authors have the option to publish the peer review history of their article (what does this mean?). If published, this will include your full peer review and any attached files.

Reviewer #1: **Yes: **Ahmed Adeel

Reviewer #2: No

Reviewer #3: **Yes: **Walid M. S. Al-Murisi

**Figure resubmission:**
---

## [Editor Report · Decision Letter 1]

22 Oct 2025

Dear DR. BER,

We are pleased to inform you that your manuscript 'Knowledge, attitudes and practices toward Female Genital Schistosomiasis among women living in communities along the Volta Lake in Volta Region, Ghana' has been provisionally accepted for publication in PLOS Neglected Tropical Diseases.

Best regards,

Angela Monica Ionica, Ph.D.

Academic Editor

Jong-Yil Chai

Section Editor

Shaden Kamhawi

co-Editor-in-Chief

Paul Brindley

co-Editor-in-Chief

---

## [Editor Report · Acceptance letter]

Dear Dr Der,

We are delighted to inform you that your manuscript, "Knowledge, attitudes and practices toward Female Genital Schistosomiasis among women living in communities along the Volta Lake in Volta Region, Ghana," has been formally accepted for publication in PLOS Neglected Tropical Diseases.

Best regards,

Shaden Kamhawi

co-Editor-in-Chief

Paul Brindley

co-Editor-in-Chief
